# The Therapeutic Relevance of Urolithins, Intestinal Metabolites of Ellagitannin-Rich Food: A Systematic Review of In Vivo Studies

**DOI:** 10.3390/nu14173494

**Published:** 2022-08-25

**Authors:** Wai-Kit Tow, Pui-Ying Chee, Usha Sundralingam, Uma Devi Palanisamy

**Affiliations:** 1School of Pharmacy, Monash University Malaysia, Jalan Lagoon Selantan, Bandar Sunway 47500, Selangor, Malaysia; 2Jeffrey Cheah School of Medicine and Health Sciences, Monash University Malaysia, Jalan Lagoon Selatan, Bandar Sunway 47500, Selangor, Malaysia

**Keywords:** urolithin, ellagitannin, pharmacological properties, mitophagy, mitochondrial biogenesis, autophagy

## Abstract

The therapeutic effects of food rich in ellagitannins have been established to stem from its microbial metabolite, urolithin. Over the past decade, there has been a growing trend in urolithin research pertaining to its pharmacological properties. The purpose of this systematic review is to collate and synthesise all available data on urolithin’s therapeutic ability, to highlight its potential as a pharmaceutical agent, and prospective direction on future research. Methods: This systematic review was written based on the PRISMA guideline and was conducted across Ovid via Embase, Ovid MEDLINE, Cochrane Central Register for Controlled Trials, and Web of Science Core Collection. Results: A total of 41 animal studies were included in this systematic review based on the appropriate keyword. The included studies highlighted the neuroprotective, anti-metabolic disorder activity, nephroprotective, myocardial protective, anti-inflammatory, and musculoskeletal protection of urolithin A, B, and its synthetic analogue methylated urolithin A. The Sirt1, AMPK, and PI3K/AKT/mTOR signalling pathways were reported to be involved in the initiation of autophagy and mitochondrial biogenesis by urolithin A. Conclusions: This review methodically discusses the therapeutic prospects of urolithins and provides scientific justification for the potential development of urolithin A as a potent natural mitophagy inducer for anti-ageing purposes.

## 1. Introduction

Urolithin(s) (Uro), a group of gut metabolites derived from ellagitannin (ET) and ellagic acid (EA), has gained much attention in recent years due to its various bioactive properties. ETs are commonly found in such as some fruits (pomegranate, rambutan, longan, and guava), berries (blackberries, strawberries, cloudberries, and raspberries), and nuts (chestnuts, pecans, walnuts, and almonds) [1,2]. ET and its consumption have been studied extensively in randomised controlled trials (RCT) and human clinical trials due to its therapeutic effect on various metabolic disorders and oxidative stress-associated disorders [3,4,5,6]. Noteworthily, differences exist in the metabolism of ET and EA depending on the variability in human metabotypes.

Four metabotypes have been recently described, of which metabotype A produces urolithin A (UroA) and its conjugates; metabotype B produces UroA, iso urolithin A (IsoUroA), and urolithin B (UroB); metabotype R produces urolithin AR (UroAR); and lastly, metabotype 0 is incapable of producing any Uro [7,8]. However, with long-term or high-dose intake of ET or EA, conversion from metabotype 0 into metabotype A or B is a possibility for some individuals [7]. This variation can be mostly explained by the differences in gut microbiota, consequently leading to the variation in health benefits associated with ET consumption [7,9]. Interestingly, metabotype B has been correlated with individuals with metabolic syndromes or chronic diseases secondary to gut dysbiosis. Additionally, a recent study showed that individuals with a higher Firmicutes-to-Bacteroidetes ratio (where a low ratio was associated with gut dysbiosis) and a greater abundance of *Clostridiales*, *Ruminococcaceae*, and *Akkermansia muciniphilia* are a better UroA producer [7]. To date, the bacterial strains involved in the metabolic pathway of Uro conversion still remain elusive [7,8,9,10].

Increasing evidence suggests that the bioactivity of ET stems from the Uro compounds, given that ET is poorly absorbed due to its high molecular weight and hydrophilicity [7,11]. Upon hydrolysis, ET is converted into EA by tannases, which are further transformed into various Uro by gut microbiomes. Sequentially, Uro D, C, A, and B are produced from the bacterial metabolism of EA, which has increasing lipophilicity and absorption capability [1,12,13,14]. The nomenclature of Uro is based on the hydroxyl substituents in its dibenzopyran-6-one core [7,11]. Uro(s) have been observed in systemic circulation after consumption of various ET-rich foods such as pomegranate [15,16,17], berries [16,18], and nuts [12,16,19,20], as well as their waste residues from industrial manufacturing [7,21].

In the past decade, there has been a growing amount of literature focusing on the bioactivity of Uro and a few emerging randomised controlled trials on Uro [22,23]. Presently, Zhang et al. provided an overview on the metabolic distribution, health benefits, and biotransformation of UroA [24]. The authors also highlighted the molecular mechanism of UroA in anti-inflammatory, anti-obesity, neuroprotective, anti-carcinogenic, and uric acid reduction reported in the latest literature. However, there are still insufficient data on Uro in general, as well as the anti-ageing aspects, in relation to autophagy, mitophagy, mitochondrial biogenesis modulation of UroA. Thus, this paper focuses more on the latter.

In light of these crucial findings, there has not yet been a systematic review reporting on this microflora-derived metabolite. In addition, the wide variation of doses, routes, and duration of Uro supplementation in the existing in vivo literature warrants an investigation. An overview on the possible range of therapeutic doses, administration regimen, genes influenced, and the beneficial effects of Uro, which can be further extrapolated in human trials, is, therefore, essential. Ergo, the present systematic review aims to provide an insight on the potential regimen, therapeutic effect, and the proposed mechanisms of Uro, which will further help in elucidating its effectiveness with the hope to serve as an essential guideline and structural framework for future investigation. Thus, this paper focuses more on the latter.

## 2. Methods

This systematic review was conducted following the requirements of the Preferred Reporting Items for Systematic reviews and Meta-Analyses (PRISMA) [25]. The data extraction and data synthesis were performed independently by the authors (W.-K.T.) and (P.-Y.C.) based on the PICO component (Population; Intervention; Comparison; Outcome) [26]. This work was registered with PROSPERO (No. CRD42021278111).

### 2.1. Search Strategy

After identifying the research objectives, a search was conducted across four electronic databases (Ovid via Embase; Ovid MEDLINE; Cochrane Central Register for Controlled Trials; Web of Science Core Collection) to identify relevant studies. The single keyword “Urolithin” was used as the primary and only search term. The search was limited to title and abstract. The search period was limited to articles published between January 2011 and December 2021.

### 2.2. Article Eligibility Criteria

Articles selected were based on the bioactive properties exhibited by urolithins and assessment of the efficacy of the bioactive properties exhibited by urolithins. The inclusion criteria were the following: (1) Original scientific research papers; (2) Published in English language; (3) Between the last 11 years (2011–2021); (4) In vivo (animals) studies. Exclusion criteria were the following: (1) Original research study related to in vitro, ex vivo studies (animals), and human trials; (2) Other exposures or parameters not related to pharmacological properties; (3) Any other studies without a separate control group; (4) Literature reviews, systematic reviews, meta-analysis, patents, and conference proceedings.

### 2.3. Data Extraction

Based on the inclusion and exclusion criteria listed above, abstract, title, and full-text screening were carried out independently by two reviewers (W.-K.T. and P.-Y.C.) using the Covidence software (https://www.covidence.org/, accessed on 5 December 2021). Instances of discordance were resolved by a third senior reviewer (U.S.) to prevent biases. The extracted information included in this review: (1) First name of the author; (2) Publication year; Population: (3) Animal (species, strain, sex, age, and number of animals per group); Intervention: (4) Type of urolithin; (5) Mode of administration; (6) Dosage, duration, and frequency; (7) Pharmacological properties; Comparator: (8) Non-exposed control group, vehicle group, placebo group, or groups administered with any other compound that shares similar pharmacological properties. Outcome: (9) Types of pharmacological properties and the therapeutic effectiveness of urolithin; (10) Mechanism of the therapeutic effect.

### 2.4. Quality Assessment

The methodological quality of the included studies was assessed independently by W.-K. T and P.-Y. C. through using the SYRCLE’s risk of bias tool [27]. Instances of discordance were resolved by discussion among the reviewers and, ultimately, consensus. The domain that SYRCLE’s risk of bias tool assessed includes sequence generation, baseline characteristics, allocation concealment, random housing, blinding (performance bias), random outcome assessment, blinding (detection bias), incomplete outcome data, selective outcome reporting, and other sources of bias. Grading of each study in their respective domain was classified as low risk, high risk, or unclear risk.

## 3. Results

The systematic search yielded a total of 859 articles from all four electronic databases. A total of 248 articles remained after the removal of duplicates from both Endnote and Covidence. Fifty-five articles remained post title and abstract screening. Subsequently, the full-text screening reduced the number to thirty-nine articles. Two articles were included as secondary references, resulting in a total of forty-one articles for qualitative synthesis, as depicted in Figure 1. The extracted data were collated in Appendix A.

### 3.1. Study Characteristics

Thirty-three studies used UroA [28,29,30,31,32,33,34,35,36,37,38,39,40,41,42,43,44,45,46,47,48,49,50,51,52,53,54,55,56,57,58,59,60], eight studies used UroB [52,61,62,63,64,65,66,67], and two studies used modified/synthetic analogue of UroA [33,68] as their treatment intervention. Most of the studies were conducted on mice (n = 38); strains included C57BL/6 (n = 18) [29,30,32,33,34,35,36,41,45,46,49,51,54,55,59,60,66,67], BALB/c (n = 3) [28,56,68], institute of cancer research (ICR) (n = 2) [37,38], APP/PS1 (n = 2) [43,48], DBA/2J [31], ape^-/-^ [61], BALB/cJInv [44], Nrf2^-/-^ [33], AhR^-/-^ [33], IL10^-/-^ [42], 3xTgAD [48], FUNDC1^f/f^ [50], FUNDC1^CKO^ [50], 10ScSn [53], 10ScSn-Dmd^mdx^J [53], mdx/Utr^-/-^ (n = 1) [53], and unknown (n = 1) [61]. Eleven of the studies were conducted on rat species which included Sprague Dawley (n = 8) [35,39,40,47,58,63,64,65], Wistar Kyoto (n = 2) [52,57], and Wistar Han (n = 1) [35]. Majority of the studies administered urolithin orally, as shown in Figure 2. Most of the studies were interventional (n = 42) [28,29,30,31,33,34,35,36,37,38,39,40,41,42,43,44,45,47,48,49,50,51,52,53,54,55,56,57,58,59,61,62,63,64,65,66,67,68], while the remaining were preventive (n = 6) [32,33,46,60] (Table 1).

### 3.2. Quality Assessment

The summarised quality assessment of the included studies is tabulated in Figure 3. The animals were randomly housed in all the studies, while only one study described the method of randomisation. Baseline characteristics were reported in 46% (19) of the studies, while only 51% (21) of the studies reported partial information of the baseline characteristics. Nevertheless, 7.32% (3) of the studies reported conflict of interests which might introduce other sources of bias. Allocation concealment, random housing of animals, and blinding of investigators were unclear in 97.6 to 98% of the studies.

### 3.3. Pharmacological Activities of Urolithin(s)

Urolithin, an intestinal microbiota metabolic product of ellagitannin, has been scientifically proven to exhibit numerous pharmacological properties, specifically UroA, which was the focus of most of the included studies. A total of 41 in vivo studies were carried out in animals, among which included neuroprotective (n = 6) [29,37,43,44,48,67], hepatoprotective (n = 1) [38], anti-metabolic disorder activity (n = 7) [30,31,45,51,52,54,61], nephroprotective (n = 7) [32,38,39,41,49,59,65], anti-tumour activity (n = 3) [28,62,68], myocardial protective (n = 5) [46,50,57,63,64], anti-inflammatory (n = 6) [33,34,42,55,66,67], musculoskeletal protective (n = 6) [35,36,40,47,53,58], osteogenic (n = 2) [55,56], and radioprotective (n = 1) [60] studies. It was noted that some studies reporting anti-metabolic disorder also reported nephroprotective and hepatoprotective effects.

#### 3.3.1. Neuroprotective 

Neurodegenerative disease


*Urolithin A*


Reversion of neurodegenerative disease has been shown as one of the therapeutic properties of UroA due to its immunomodulatory effects, specifically in Alzheimer’s disease animal models. The amyloid hypothesis suggested that extracellular amyloid beta (Aβ) deposition in the brain is the pathological underlying cause of Alzheimer’s disease, due to the mutation in β-amyloid precursor protein (APP) gene [69,70]. Apart from that, Apolipoprotein E4 (APOE4), a variant of apolipoprotein, was also known to be a risk factor of Alzheimer’s disease due to their ineffectiveness in promoting the proteolytic breakdown of Aβ [71]. UroA was shown to significantly improve cognitive function in APP/PS1 mice compared to the vehicle-treated group when assessed using Morris water maze test, at dosages of 300 mg/kg and 200 mg/kg [43,48]. Significant reduction in the loss of NeuN^+^ immunoreactivity (CA1 region of hippocampus) and cellular apoptosis (cortex and hippocampal CA1) through UroA treatment was also demonstrated by Gong et al. [43]. In relation to neurogenesis, Gong et al. reported significant increase of bromodeoxyuridine-positive (BrdU^+^) cells, as well as doublecortin-positive (DXC^+^) cells (dentate gyri) in the UroA-treated group [43]. Both studies reported a significantly smaller mean area and density of Aβ plaque deposit (Aβ)40-positive and Aβ42-positive) in the UroA-treated group compared to the vehicle-treated group [43,48]. Similar results were found in soluble Aβ40 and Aβ42 in the cortex hippocampus [43]. Reactive microglia and astrocytes around Aβ plaques were also reduced in the brains of the UroA-treated group. The UroA-treated group also showed significantly less intense reactive astrogliosis and microgliosis compared to the vehicle-treated group. In addition, the researchers observed that UroA treatment showed a remarkable increase in the expression of p-AMPK and significant decrease in the expression of p-P65NF_K_B, p-P38MAPK, Bace1, and APP. This resulted in the decrease of the Aβ production and the facilitation of lysosomal degradation of Aβ [43,72]. Similar finding was reported by Fang et al. (2019) on the increased shifting of microglial towards phagocytic state and increased engulfment of Aβ plaques by microglia, consistent with the findings of Gong et al. (2019) in the facilitation of lysosomal degradation [43,48]. Another pathway identified was through the activation of AMPK, resulting in the reduction of NF_K_B signalling by activating SIRT1 and decreasing the expression of proteins associated with apoptosis [43]. UroA was also found to be effective in increasing the maximal OCR of APOE4 neurons, comparable with that of wild-type mice control model [48].

A separate study conducted by Chen et al. (2019) examined the effect of UroA with the cholinergic hypothesis as the pathological cause of Alzheimer’s disease. The study postulated the disease was a result of reduced synthesis or increased destruction of acetylcholine [37,73]. The team studied the effects of UroA at three levels of dosage, from high (150 mg/kg), medium (100 mg/kg), to low (50 mg/kg) in the D-gal (D-galactose)-induced brain ageing model. [37]. Significant increase in brain index was noted in the UroA-treated group. UroA treatment also significantly lowered acetylcholinesterase (AChE) and monoamine oxidase (MAO) in the whole brain of the ageing model group compared to the untreated group. Furthermore, UroA-treatment was found to significantly reverse the decrease in catalase (CAT), glutathione peroxidase (GSH-Px), superoxide dismutase (SOD), and total anti-oxidant capacity (T-AOC) activities and the increase in malondialdehyde (MDA) levels in brain tissues of the model group. UroA-treatment was found to attenuate the histopathological changes, such as pyramidal cells, karyopyknosis, disorganised nerve fibres with irregular neurons, and expanded gap of neurons that were observed in the CA3 region of the model mice group. Specifically, a high dose of UroA (150 mg/kg) was able to provide maximum protective effect with only minute changes observed in the morphological structure of brain tissues. The significantly decreased number of neurons and Nissl bodies in the hippocampus of the model group were recovered in the UroA-treatment group. Their findings also showed UroA to be associated with the regulation of SIRT1 and p53 phosphorylation in the model group. This was observed in a concentration-dependent manner in decreasing the levels of p53, p21, and miR-34a, as well as the increased expression of SIRT1 in UroA treatment [37]. The treatment with UroA also significantly downregulated the glial fibrillary acidic protein (GFAP) expression and significantly decreased the number of GFAP immunoreactive astrocytes in mice of the model group. However, studies by Fang et al. (2019) reported no changes in the GFAP expression in Alzheimer’s disease animal model [48]. This might be due to the aetiology of the animal disease model that Fang et al. (APP/PS1 mice) and Chen et al. (D-gal-induced ICR mice) used for their studies. Moreover, these four studies reported homogenous evidence of the immunomodulatory effect of UroA in suppressing the proinflammatory factors (TNF-α, IL-1β, IL-23, and IL-6) in the brain, either through the inhibition of Th17 cell differentiation or regulation of the expression of genes and proteins associated with autophagy, as well as apoptosis [29,37,43,48]. These immunomodulatory effects showed that UroA was able to exert its protective effect on neurodegenerative diseases.


*Urolithin B*


Another type of urolithin, UroB, also showed promising results in exhibiting a neuroprotective effect in D-gal-induced ageing C57BL/6 mice model. UroB was able to ameliorate the effect of D-gal-induced impaired learning ability and memory performance, improve behaviour impairment and cognitive disorder, and induce hippocampal long-term potentiation (LTP), as reported by Chen et al. (2021). Moreover, UroB was reported to exert anti-oxidant properties that increased the level of biochemical indicators, such as GSH-Px, SOD, CAT, and T-AOC, in the brain tissue of the D-gal-induced mice. On the contrary, MDA was significantly reduced in UroB treatment compared to the D-gal-induced ageing control mice model. In addition, UroB at the dosage of 150 mg/kg was also capable of significantly promoting the activities of anti-oxidant enzymes, such as Cu, Zn-SOD, and CAT [67].

The histopathological analysis showed that UroB administration reversed structural damages observed in the UroB-treated D-gal-induced mice model. Structural damages, such as disorganised nerve fibres with irregular neurons and apoptosis-like-cells, were found in the hippocampal of D-gal-induced mice model group. More importantly, the 150 mg/kg dose of UroB treatment exerted the maximum protective effect, equivalent to the results of the control group. UroB prominently suppressed the activation of glial cells and exerted a protective effect in improving hippocampal pathology in D-gal-induced mice. The UroB intervention significantly reduced the formation of advanced glycation end-products (AGEs), carboxymethyl lysine (CML), carboxyethyl lysine (CEL), and the expression of receptor for advanced glycation end-products (RAGE) [67].

The total number of TUNEL-positive cells was remarkably reduced in the UroB-treated group. The expression of Bax was decreased while the expression of Bcl-2 was increased, indicating its inhibitory effect on neuron apoptosis. Furthermore, Chen et al. reported UroB increases the expression of p-JNK, p-p38 MAPK, and pAkt and phosphorylation of p44/42 MAPK, and resulted in a noticeable corresponding decrease in the activity and release of Cytochrome C (Cyt C) from mitochondria into the cytosol. The team investigated the effect of UroB treatment on Bcl-2-associated death promoter (Bad), a proapoptotic protein of the BLC2 family when phosphorylated. UroB was reported to significantly promote the phosphorylation of Bad in the study [67].

Next, Chen et al. conducted a comparison study on the effect of UroB on the normal young mice model and naturally senile mice model. UroB was reported a decrease in mRNA levels of AGEs, CML, CEWL, and RAGE in both groups, while an increase in the expressions of Cu, Zn-SOD, and CAT in the naturally senile mice model compared to the normal young mice. UroB was revealed to significantly increase the Cyt C in mitochondria in the naturally senile mice mode, while markedly abolishing accumulation of cytosolic Cyt C in normal young mice model. Other findings include downregulated JKN/p38, upregulated PI3K/Akt, improved cognitive deficits, amelioration of structural plasticity, suppression of apoptosis, attenuation of astrocyte, and microglial activation in hippocampal tissue of normal ageing mice [67].

Autoimmune disease


*Urolithin A*


Using the experimental model of autoimmune encephalomyelitis (EAE), Shen et al. (2021) demonstrated the neuroprotective effect of UroA at a dosage of 25/mg/kg/day [29]. UroA was able to inhibit the progression of EAE. Shen et al. also conducted immunohistological analysis on the lumbosacral enlarged spinal cord sample of the animal model and found pronounced inflammatory infiltration and demyelination of the white matter in the vehicle-treated group [29]. On the other hand, the UroA-treated group exhibited a significantly increased portion of intact myelin in the spinal cord. In addition, the team studied the effects of UroA treatment on dendritic cells (DCs) and CNS mononuclear cells (MNCs). The proportion of CD11c^+^ DCs infiltration into the CNS in the UroA-treated group was significantly lower compared to the vehicle-treated group. Similarly, the proportion of co-stimulatory molecules CD80, CD86, CD40, and CD14 expressed on CD11c^+^ were remarkably reduced in the UroA-treated group. The absolute number of infiltrating cells (infiltrating macrophages, activated microglia, microglia, and M1-type microglia) were also significantly reduced. In regard to the migration of pathogenic T cells from the periphery to CNS, the UroA-treated group showed remarkable decrease in the percentages of CD45^+^, CD3^+^, CD4^+^, and CD8^+^ cells infiltrating the CNS. Th1 and Th17 cells were also found in reduced amounts, without any significant differences between the UroA- and vehicle-treated groups. The evidence showed that the suppression of the immune response by UroA at an early stage of EAE in this study was most likely attributed to (1) the inhibition of Th17 cell differentiation [29], (2) acting as a ligand molecule and directly targeting aryl hydrocarbon receptor (AhR) [29], (3) the upregulation of the inhibitory factor IL-10 [29,48], and (4) the reduction of the expression of inflammatory factors necessary for the differentiation of Th17 cells [29].

Opportunistic infection


*Urolithin A*


In the application of mitigating cerebral toxoplasmosis, Tan et al. (2020) demonstrated the neuroprotective effects of UroA at a dosage of 30 µg among BALB/cJInv mice for 39 days [44]. *Toxoplasma gondii*-infected mice groups treated with UroA survived through the entire experimental duration, while 40% of the infected DMSO control mice succumbed to acute infection and died 10 days post-infection. Cyst size in UroA-treated mice had significantly smaller cyst diameter and mild reduction in cyst load compared to the DMSO control mice. This could be due to the inhibitory effect of UroA on cyst formation in the brain. When assessed for the innate response of infected mice towards predatory cat odour, the UroA-treatment group was less willing to leave the bisector containing their odour and spent less time in the cat odour bisector as compared to the DMSO control group. Thus, the study postulated a possibility of correlation between the decreased severity of cyst load due to the UroA administration and increased perceived risk of the infected mice towards cat odour [44].

#### 3.3.2. Hepatoprotective


*Urolithin A*


The ameliorative effect of UroA on liver damage in an ageing mice model was demonstrated by Chen et al. (2020) [38]. UroA was able to significantly improve the activities of SOD, GSH-Px, CAT, and T-AOC in the liver, while lowering the MDA level compared with the ageing model group. In regards to its immunomodulatory effect, UroA also exhibited effects in the liver by significantly lowering the mRNA expression of TNF-α, IL-1β, and IL-6. Nonetheless, UroA treatment was also reported to restore the hepatic functions, indicated by the decreased levels of ALT and AST [38]. Furthermore, it was also reported to elevate mitochondrial biogenesis in the liver, evident by the significant elevation of mtDNA/nDNA ratio in UroA-treated high fat diet animal model group [30]. Thus, it is postulated that this could improve the metabolic profiles of the animal model group.

#### 3.3.3. Nephroprotective


*Urolithin A*


Autophagy is known as a cell-protective mechanism in kidney ischemia-reperfusion injury (IRI). Wang et al. (2019) demonstrated that UroA, a strong autophagy inducer, is protective against acute kidney injury. The study used the IRI model pretreatment of UroA (50 mg/kg), three days prior and 30 min prior to the IRI induction surgery, which resulted in a significant inducement of transcription factor EB (TFEB) nuclear localisation [32]. This indicated an increase in autophagy and lysosomal biogenesis, which is protective against IRI. Besides, UroA pre-treatment also led to a significant reduction of kidney injury markers, such as BUN, NGAL, creatinine, and KIM-1, and inflammatory markers including TNFα, IL1β, MIP1α, and MIP2 mRNA levels. This implies that UroA plays a role in the reduction of oxidation and inflammation, and modulation of autophagy. The optimal dose of 50 mg/kg pre-treatment was also supported by Zhang et al. (2021), with a similar positive result in attenuating kidney damage and oxidative stress, and a significant reduction in TUNEL-positive cells and Cleaved caspase-3 was exhibited. The autophagy modulation and anti-oxidative effect of UroA was proposed to be mechanistically linked to the p62-Keap 1-Nrf2 pathway following the observations of significant increase in Nrf2 level and a reduction of Keap 1 and p62 expressions [59]. Interestingly, the study exhibited the nephroprotective effects to be dose independent, where a higher dose (100 mg/kg/day) showed a counteractive response with an increase in TUNEL-positive cells and cleaved caspase-3 and had a comparable anti-oxidative effect to the 50 mg/kg/day dose. This implies an underlying mechanism that is yet to be known.

The anti-apoptotic, anti-oxidative, and anti-inflammatory properties of UroA has shown to be beneficial against cisplatin-induced nephrotoxicity by several studies [39,41,49]. Cisplatin is a common anti-cancer drug used in combating malignancy, including in the lungs, ovaries, soft tissues, bones, and muscles. However, it is often associated with nephrotoxicity that requires dose reduction or withdrawal due to its propensity to accumulate in the kidney, and its potent apoptotic, inflammatory and oxidative effect [74,75]. The study by Guada et al. (2017) and Jing et al. (2019) showed similar protective effects in rats (50 mg/kg) and mice (100 mg/kg) after UroA pre-treatment. Example of protective indicators include the significant attenuation of creatinine levels, reduction of tubular damages and proinflammatory biomarkers including TNF-a and MIP2, as well as reduction of immune cell infiltration [39,41]. Anti-oxidant effect was also seen via the significant reduction of protein nitration and lipid peroxidation and increase in glutathione peroxidase content and SOD activity [41]. Although only a considerable decrease in tubular apoptotic cells was observed by Guada et al. (2017), a significant modulation of apoptotic marker (Caspase-3) and DNA fragmentation was reported by Jing et al. (2019). This indicates the presence of a minor beneficial anti-apoptotic effect [39,41]. Interesting results were observed by Zou et al. (2019) that subjected mice to an oral dose of 50 mg/kg of P2Ns-GA-encapsulated Urolithin A, three times a week for up to 19 days [49]. The nanoparticles delivery method which targeted the gut-expressed transferrin receptor was shown to increase the oral bioavailability by seven-fold compared to the non-encapsulated UroA. The results also demonstrated a decreased mortality rate by 63% due to the anti-oxidative and anti-apoptotic effects as evident from the reduction of nuclear factor erythroid 2-related factor 2- and P53-inducible genes.

Lastly, UroA was proven to be protective against kidney ageing, specifically kidney cell senescence as elucidated by Chen et al. (2020) [38]. Kidney cell senescence is mainly caused by telomere attrition as well as stressors such as DNA damages and oxidative stress [49]. Using a D-galactose induced ageing mice model, Chen et al. (2020) found a beneficial dose-dependent effect of UroA on reversing kidney damage especially with the dose of 150 mg/kg. Observation of UroA treatment showed better histopathological and kidney function preservation compared to untreated ageing model. The study also exhibited significant effects of UroA treatment on increasing anti-oxidants levels and total anti-oxidant capacity, and downregulation of inflammatory markers including TNF-a, IL-1b, and IL-6, in addition to the modulation of apoptotic markers such as Bax, Bcl-2, and cleaved caspase-3. Thus, the strong anti-oxidative, anti-inflammatory, and anti-apoptotic properties of UroA was proven to play a potential role in combating kidney ageing.


*Urolithin B*


Li et al. (2020) studied the effects of UroB against renal fibrosis in hope of finding potential therapy in chronic renal failure [65]. Promising results were obtained especially with high doses of oral UroB (80 mg/kg/day) treatment. Using a unilateral ureteral obstruction (UUO) rat model, a high dose of UroB consistently showed significant nephroprotective effects via the preservation of renal function proven from decreased proteinuria, Cr, and BUN. Additionally, a marked reduction in fibrotic area and pro-inflammatory factors, including TNF-α, IL-6, and monocyte chemoattractant protein-1 (MCP-1), were observed in all UroB-treated groups. The team subsequently concluded that the nephroprotective effect of UroB was mechanistically linked to the regulation of the TGF-β1/Smad and TLR4/NF-κB pathway. They also observed reduction of the related mediators, such as TGF-β1, Smad2, and Smad3 proteins, as well as TLR4, NF-κB p65, p-IKKα, and TRAF6 in UroB-treated groups.

#### 3.3.4. Anti-Metabolic Disorder Activity


*Urolithin A*


UroA showed promising results in improving systemic insulin sensitivity, attenuating triglyceride accumulation, and improving serum lipid profile. UroA has also shown to reduce adipocyte hypertrophy and macrophage infiltration into the adipose tissue, and alter M1/M2 polarisation in peritoneal macrophages [30,31,45,51,52]. The metabolic profile improvement was augmented by the enhanced mitochondrial biogenesis in the liver and adipose tissue of UroA through the promotion of macrophage M2 differentiation [30]. A study conducted by Yang et al. (2020) revealed that UroA is a potent blood glucose (decrease), adiponectin (increase), and mitochondrial dynamic (increase) regulator compared to EA [31]. Markers of mitochondrial function were noted to significantly increase in both liver and skeletal muscle (Mfn2, Prkn, and Pink1), except for the markers in epididymal fat, and Errα and Pgc1α expression [31]. Mitochondrial density of the skeletal muscle and liver of the UroA supplemented mice group were significantly increased. Similar findings on the modulation of the cleaved-caspase 3 and cleaved-caspase 1 expression in the brain was also reported by Jing et al. (2019), Zhang et al. (2021), and Tuohetaerbaike et al. (2020) [41,45,51].

In 2020, Tuohetaerbaike and his team demonstrated the pancreas-protective effects of UroA (50 mg/kg/d) against type 2 diabetic mice models [51]. The diabetic mice model fed with UroA showed significantly decreased fasting blood glucose, after-glucose-loading glucose, glycated haemoglobin levels, plasma C-peptide, MDA, TNF-α, and IL-1β. Moreover, the UroA-treated mice model also showed increased GSH, IL-10 content, and glucose tolerance. The pancreas protective effect of UroA was evident in the improved pathological and morphological features of the pancreas. These improvements were accompanied by the upregulation of the protein levels of microtubule-associated protein 1 light chain 3-II (LC3II) and beclin1, and the downregulation of sequestosome 1 (p62). As reported by Tuohetaerbaike et al., the protective effect was partially mediated by the regulation of autophagy and AKT/mTOR signal pathway by UroA [51]. Abdulrahman et al. (2021) reported significant decrease in body weight and restored serum lipid profile in high-fat diet (HFD)-induced obesity rats model treated with UroA. Significant decrease in serum cholesterol, triglycerides, and LDL-C levels, and a significant increase in HDL-C level compared to the HFD model group was also observed [52].

In addition, a similar study on high-fat diet-induced obesity mice model by Zhang et al. (2021) focused on the effect of UroA in pancreatic inflammation. The study revealed that UroA treatment significantly suppressed TXNIP protein level, NLRP3 inflammasome, IL-1β, and TNF-α levels, while it increased the IL-10 level in the pancreas of the HFD mice model group. TXNIP is a critical signalling protein that mediates inflammation and pancreatic β cell death, while NLRP3 inflammasome is responsible for the trigger of islet inflammation in type 2 diabetes [76,77]. UroA was observed to inhibit endoplasmic reticulum (ER) glucolipotoxicity-induced TXNIP-mediated NLRP3/IL-1β inflammation signals, indicating that UroA inhibited pancreatic inflammation in the high-fat diet-induced mice model group [54].


*Urolithin B*


Zhao et al. (2019) reported the ability of UroB in decreasing lipid plaque deposition in vivo at a dosage of 10 mg/kg per day, while reversing cholesterol transport pathway in vitro through influencing certain key proteins in the apoE^-/-^ mice model [61]. The reduction of lipid plaque was evidently shown through the reduced bright red punctate via Oil Red O staining of the aorta. An in vitro study showed that the expression of key proteins influenced by UroB in reversing the cholesterol transport pathway were noted to be SR-BI and ABCA1. In addition, UroB was able to increase cholesterol efflux from cholesterol laden macrophages to high-density lipoprotein (HDL) particles. Abdulrahman et al. (2021) also reported UroB to regulate dysfunctional lipid metabolism evident by the significant body weight reduction and the ability in restoring serum lipid profile [52].

#### 3.3.5. Anti-Tumour Activity


*Urolithin A*


In a study conducted by Dahiya et al. (2018), the investigation on the UroA suppressed the growth of C4-2B and PC-3 tumours on the xenograft study in BALB/c athymic mice model [28]. However, the inhibition was less significant in PC-3 compared to C4-2B tumours. The study also showed UroA to be effective in suppressing tumour markers such as Ki-67 and pAKT in C4-2B tumour xenografted mice model, while PC-3 tumour xenografted mice model only showed significant reduction in the expression of Ki67, but not pAKT, compared to the vehicle treatment (sunflower oil). The investigation demonstrated UroA to inhibit androgen receptor signalling axis to inhibit tumour growth of AR^+^ CEVEC’s amniocyte production (CaP).


*Urolithin B*


Lv et al. (2019) demonstrated the anti-proliferative effects of UroB in subcutaneous xenograft mice modelled with hepatocellular carcinoma (HCC) cells. Comparable to the findings of UroA, the researchers reported significant reduction of the cell proliferation marker Ki-67 in the UroB-treated mice model, while also reducing the size of the tumour growth compared to the vehicle-treated group. Furthermore, UroB was able to increase the phosphorylated β-catenin expression, thus effectively blocking its translocation from nuclear to cytoplasm. This resulted in the inactivation of the Wnt/β-catenin signalling pathway [62].


*Methylated Urolithin A*


In addition to UroA and UroB, a modified urolithin known as methylated UroA (mUroA) was investigated for its effect on tumour suppression by Zhou et al. (2016) [68]. In this study, the human prostate cancer DU145 cells were xenografted onto the BALB/c mice model. Four weeks of treatment with mUroA at a dosage of 20, 40, and 80 mg/kg significantly reduced the mean tumour volume to 600, 480, and 410 mm^3^, respectively, compared to the control (800 mm^3^). Moreover, miR-21 expression was found to be suppressed by mUroA treatment compared to the control group.

#### 3.3.6. Myocardial Protective


*Urolithin A*


UroA has also been proven to be beneficial for the heart. In particular, the work of Tang et al. (2017) showed that UroA helped in reducing myocardial ischemia-reperfusion (I/R) injury [46]. Myocardial I/R injury involved the injury secondary to impaired cardiac tissue oxygen supply as well as the injury induced from blood flow restoration [78]. Reperfusion injuries are often overlooked and results in either reversible damages, namely, reperfusion-induced arrhythmias and myocardial stunning; and irreversible damages such as microvascular obstruction and lethal myocardial reperfusion injury [79]. Some important causative factors of I/R injury include oxidative stress, intracellular calcium overload, rapid pH correction, mitochondrial PTP opening, inflammation, and apoptosis [78,79]. According to Tang et al. (2017), the administration of UroA (1 mg/kg) intraperitoneally in male mice 24 h and one hour before ischemic induction significantly reduced infarct size, while TUNEL assay analysis revealed significant reduction in cardiomyocyte apoptosis. Significant improvement was also observed in left ventricular function and ejection fraction in the UroA pre-treated group and prevention of the increase in CK and LDH secondary to I/R injury.

UroA is also shown to be protective against septic cardiomyopathy as demonstrated by Wang et al. (2021) [50]. It is a reversible myocardial depression condition secondary to septic shock, where inflammation and mitochondrial dysfunction plays a role [80]. Pre-treatment with of UroA (30 mg/kg) administered via intraperitoneal route in FUNDC1*^f/f^* mice prior to sepsis induction markedly suppressed myocardial injury biomarkers including (LDH, troponin T and CK-MB). It also normalised cardiac function (left ventricular ejection fraction, left ventricular diastolic dimension and fractional shortening) associated with sepsis. Noteworthily, in FUNDC1 knockout mice (FUNDC1*^CKO^*), UroA was unable to produce any beneficial effect, suggesting the involvement of FUNDC1-associated mitophagy in the cardioprotective effect.

A recent study by Albasher et al. (2022) showed UroA to prevent diabetic cardiomyopathy in rats [57]. Diabetic cardiomyopathy is a diabetic-associated heart failure that is not caused by other cardiac risk factors [81]. The condition is associated with left ventricular hypertrophy, inflammation, fibrosis, and apoptosis [57,81]. With a dose of 2.5 mg/kg of UroA, the study reported a significant improvement in rats’ cardiac function and reduction in cardiac markers (Troponin-1 and creatine kinase) after 8 weeks of intraperitoneal treatment as compared to the untreated diabetic mice group. The UroA-treated group also significantly reversed the pathological condition in the diabetic group including a reduction in collagen deposition and inflammation whilst promoting anti-oxidative and anti-apoptotic effect. SIRT1 signalling was proposed to be the underlying mechanism of UroA’s protective effect secondary to the increase in mRNA and total/nuclear protein levels of SIRT1. This also resulted in the reduction in acetylation of FOXO1, Nrf2, NF-jB, and p53 levels in addition to the reversal of all UroA cardioprotective effects after administration of an SIRT1 inhibitor called Ex-527.


*Urolithin B*


UroB also possesses some cardioprotective effects, as evidenced by two studies. A study by Gao et al. (2020) reported UroB to be protective against post-myocardial infarction (post-MI) ventricular remodelling and arrhythmia in rats induced myocardial infarction (MI) after 2 weeks of intraperitoneal administration of UroB [63]. Although post-MI arrhythmias are mostly benign and self-limited, a rough estimation of around 90% of patients has some form of arrhythmic complication during or post-MI especially after an ST-elevation MI, and some may even led to hemodynamic compromise [82]. Gao et al. (2020) revealed that the higher dose (5 mg/kg/day) was more effective compared to the lower dose (2.5 mg/kg/day), as only the higher dose alleviated the decline in cardiac function indexes. However, both doses were similarly effective in increasing dP/dt max and in reducing brain natriuretic peptide levels. Further investigation utilising the higher dose of UroB showed a smaller infarct area and diminished post-MI cardiac remodelling effect, such as collagen deposition and fibrosis, as evident from the staining and mRNA expression. Significant anti-inflammatory effect of UroB was demonstrated via the suppression of CD68^+^ macrophage infiltration and proinflammatory markers (TNF-α and IL-6). Furthermore, UroB prevented the heart rate increase and P wave prolongation, as found in the MI group. It also reduced the arrhythmia inducibility post-MI compared to the untreated MI group (20% vs. 66.7%). The UroB mechanism was proposed to be related to the JAK/STAT3 and Smad2/3 signalling pathway. Inactivation of phosphorylation of JAK2/STAT 3 was observed, where activation was shown to be related to cardiac dysfunction via inflammation and fibrosis [83]. Thus, it is postulated that the UroB effect in alleviation of post-MI arrhythmia is mediated by the effect of reduction in inflammation and fibrosis. Cardiac arrhythmia is rather unpredictable and dysfunction can be attributed to cardiac remodelling following events such as myocardial injury, ageing, and inflammation [84]. The ability of UroB to reverse some of these processes strengthens the evidence that it could be a potential novel therapeutic agent.

Similar to UroA, as a pre-treatment, UroB also conferred some protection against myocardial I/R injury. In the experiment by Zheng et al. (2020), UroB (0.7 mg/kg) was administered intraperitoneally to 7 week-old rats as a prophylactic therapy 0, 24, and 48 h before induced I/R injury [64]. It was found that UroB had no significant effect on area at risk, but it helped to reduce infarct size from 41.5% in the I/R group to 9.3% in the UroB-treated group. UroB also lowered the biomarkers of acute MI (the serum CK and LDH level). It showed an anti-apoptotic effect as evident in reduced TUNEL-positive cells and cleaved caspase-3. A surprising 50% reduction of intracellular superoxide anion radicals post UroB pre-treatment was also observed in the area at risk. The UroB anti-oxidant effect on the heart was also strengthened by the reduction of lipid peroxidation product MDA level and the restoration of anti-oxidant enzyme SOD level. Anti-autophagy effect specifically via reduction of LC3 II/I ratio, accumulation of p62 levels, and downregulation of mTOR/ULK 1 pathway was postulated as the mechanism involved in UroB cardioprotective effect. Overall, it proposed that the anti-apoptotic and anti-oxidation properties of UroB with a mechanistic link to the p62/Keap 1/Nrf3 pathway was beneficial to combat the IR injury [64].

#### 3.3.7. Anti-Inflammatory

The pathological causes of genetic and degenerative disorders are often attributed to the mutation in the genome that causes inflammatory abnormalities. Urolithin(s) have been reported to be a potent anti-inflammatory agent in all the studies that examined its anti-inflammatory potential in relation to other diseases [28,29,30,31,32,35,36,37,38,39,40,41,43,44,45,46,47,48,49,50,51,52,53,61,62,63,64,65,68]. Specifically, there were six in vivo studies that examined its anti-inflammatory effect [33,34,42,55,66,67].


*Urolithin A*


A study by Saha et al. (2016) revealed that UroA (40 mg/kg) was able to significantly reduce phorbol myristate acetate (PMA)-induced superoxide generation in neutrophils of C57BL/6 mice. UroA-treated mice showed marked reduction in ear oedema by 47.5%, compared to its vehicle control. In addition, the extent of protection and anti-inflammatory effect against PMA-induced ear oedema of UroA, was comparable to indomethacin. Furthermore, UroA was also reported to significantly inhibit myeloperoxidase (MPO) activity. Meanwhile, the H&E staining showed reduced swelling and decreased inflammatory cells in the UroA treatment mice model [34].

According to a study conducted by Mousavi et al. (2021), UroA was able to improve the clinical outcome of acute campylobacteriosis while exhibiting immunomodulatory effect during the course of infection. UroA was administered at a dosage of 0.114 mg/kg/day in the IL10^-/-^ C57BL/6j mice model. The UroA-treated group was reported to show lower numbers of apoptotic epithelial cells counts, colonic macrophages, monocytes, and T lymphocytes compared to the placebo-treated group on day 6 post infection. Nonetheless, this was accompanied by less pronounced histopathological changes as well. IFN-γ in the colonic and ileal explants were found in lower concentrations in the UroA-treated group compared to the placebo-treated group. While TNF-α was also found in lower concentrations in both colonic and ileal explants, there was no increase of IFN-γ concentrations in the lungs of the UroA-treated group compared to the placebo-treated group. However, elevation of IFN-γ was found in the liver and kidney samples of both mice groups. The study also reported no increase in MCP-1 and nitric oxide concentrations in the ileum of the UroA-treated group. It is noteworthy that longer and large intestine lengths were reported in the UroA-treated group, indicating its intestinal protective functions [42].

Furthermore, in 2019, Singh et al. reported that by using UroA as a treatment on LPS-induced peritonitis in the C57BL/6 mice model, it significantly reduces the serum IL-6 and TNF-α levels. In another study, Tao et al. (2021) reported the immunomodulatory effects of UroA in ovariectomy-induced osteoporosis mice model [55]. The team reported markedly downregulated TNF-α, IL-1β, and IL-6 levels in the 20 mg/kg UroA-treated group compared to 10 mg/kg UroA-treated and ovariectomised (OVX) control group. On the contrary, IL-10 was reported to steadily increase in both 20 mg/kg and 10 mg/kg UroA treatment groups compared to the OVX control mice group. Thus, these three studies demonstrated the anti-inflammatory effect of UroA, most notably via the inflammatory cytokines.


*Urolithin B*


Chen et al. (2021) reported UroB treatment in the D-gal-induced mice model group to suppress the activities of inflammatory cytokines IL6, TNF-α, and IL-1β [67]. A separate in vivo study conducted by the same author reported UroB treatment to have restored the decreased IL-4 level induced by D-gal administration [66]. Besides that, the levels of proinflammatory cytokines IL-6, TNF-α, IFN-γ, and IL-1β were also found to have increased in the UroB supplemented D-gal-induced mice model group. IgA and sIgA levels in the serum and small intestine were significantly decreased in the UroB supplementation of D-gal-induced mice model group. UroB was reported to decrease MDA level, and increase CAT, SOD, T-AOC, and GSH-Px levels in the D-gal ageing mice model. Similarly, AGE content in serum and small intestine was suppressed by the UroB intervention. Protein expression of TLR4, IRAK4, TRAK6, IKKβ, NF-_K_bp65, and HMGB1 were markedly suppressed by UroB administration. Interestingly, identical results were reported in the natural ageing mice model at a dosage of 150 mg/kg/d.


*UAS03*


In 2019, Singh et al. studied a synthetic analogue derived from UroA, known as ‘UAS03’. This synthetic analogue of UroA showed potent anti-inflammatory activities compared to UroA. It was shown that UroA shared similar findings as UAS03 in the treatment of *E. coli*-derived lipopolysaccharides (LPS)-induced peritonitis [33]. The study reported significant reduction in the serum level of IL-6 and TNF-α of LPS-induced, TNBS-induced colitis, and DSS-induced mice group. Nonetheless, it is imperative to take note that UroA or UAS03 mediated protection against colitis requires AhR-Nrf2 pathways. UroA or UAS03 treatment on TNBS-induced colitis Nrf2^-/-^ mice group only manage to partially reduce serum inflammatory mediators while it failed to restore body weight loss, protection from shortening of colons, and enhancement of gut barrier integrity compared to the wild-type mice. Similar findings were also reported in the AhR^-/-^ mice group treated with UroA or UAS03 compared to the wild-type mice group [33].

#### 3.3.8. Musculoskeletal Protective

The pharmacological properties of UroA in musculoskeletal protection can be described based on two distinctive pathological causes, mainly ageing-associated diseases and genetic disorders.

Ageing-associated diseases

Skeletal muscle dysfunction is one of the indicators of ageing. The main contributing factor is the reduced ability to eliminate dysfunctional mitochondria via mitophagy leading to age-associated skeletal muscle decline: sarcopenia [85,86]. As such, the ability to improve mitochondrial function is a sought-after approach to increase muscle health among the elderly population to reduce the risk of fragility. Study conducted by Ryu et al. (2016), both acute and chronic UA supplementation irrespective of the diet, lead to an improvement in muscle quality of mouse models without affecting their body weight [35]. Specifically, the chronic administration of 50 mg/kg/day UA for 8 months in 16-month-old C57BL/6J mice fed with high fat diet showed a 9% greater grip and 57% greater level of spontaneous exercise, whilst acute administration of 50 mg/kg/day for 6 weeks in 22-month-old C57BL/6J mice with a normal chow diet showed a 42% increase in running endurance. Autophagy and specifically mitophagy was proposed as the mechanism involved through the observation of the increase in the LC3-II to LC3-I ratio, lower p62/SQSTM levels, and higher gene expression (*p* < 0.001) of mitophagy (Park2) and autophagy (Pik3c3) in UroA-treated (25 mg/kg/d) 5.5-week-old male Wistar Han rat model. Besides the reduction in mitophagy, ageing caused a reduction in skeletal muscle function via the declining levels of ATP and NAD+ and the reduction in vascular supply [36]. This phenomenon can be reversed, as observed by Ghosh et al. (2020), where after 16 weeks of 10 mg/kg UA supplementation in 12-week-old C57BL/6 mice, there was an increase in ATP and NAD+ levels and an increase in angiogenesis via the SIRT1-PGC-1α pathway [36]. The increase in angiogenesis was proven from the significant upregulation of angiogenic markers including VEGFA and CDH5 as compared to the placebo, as well as the increased expression of PGC-1α protein and SIRT1, which helps to promote angiogenesis.

The mitophagy activity of UroA has also been shown to contribute to the protection against lower back pain caused by intervertebral disc degeneration (IDD) [87]. Aside from genetic inheritance, nutritional deficiency, and a loading history, ageing is also one of the main precipitators of degenerative disc degeneration, leading to physical disruption, such as pain after minor incidents [88]. As current treatment focuses on symptomatic alleviation, the ability of UroA in preventing disease progression can be a novel therapeutic approach against this poorly treated disease that significantly reduces the quality of life [89]. The effectiveness of UroA in IDD is elucidated by Lin et al. (2020). A 4-week dietary supplementation of 25 mg/kg/day UroA dissolved in DMSO in 8-week-old rats with induced IDD showed a higher T-2 weighted signal intensity and lower Pfirrmann grade compared to the IDD (control) group. Additionally, significant reduced disc structure disruption and fibrosis of nucleus pulposus cells and partial retainment of proteoglycan matrix in UroA-treated IDD group was observed compared to its control group [87]. Apoptosis of intervertebral disc cells in the UroA-treated group was also significantly reduced compared to the IDD control group. Mitophagy upregulation via AMPK pathway activation was the proposed mechanism involved as all beneficial effects were reversed by an inhibitor of AMPK, namely, the compound C + UroA-treated IDD group. Noteworthily, a similar observation, including insignificant disc space narrowing, marked increase in T2-weighted signal intensity, significantly lower Pfirrmann grade, and increased proteoglycan and collagen, was reported by Liu et al. (2018) using similar doses [47]. The observation is supported by Shi et al., (2021), who also reported a significantly higher collagen type II and aggrecan level after 4 week of UroA therapy compared to the IDD control group [58]. Overall, the current in vivo evidence supports UroA beneficial properties against back pain secondary to IDD. Nevertheless, further clinical trials could be performed to elucidate its underlying mechanism and effectiveness.

Genetic disorders

UroA also showed beneficial effects in treating Duchenne muscle dystrophy (DMD), a type of genetic muscular dysfunction in children in the absence of dystrophin protein [53,90]. The study by Luan et al. (2021) reported that defective mitophagy was also a marker of DMD, in which UroA was found to help restore mitophagy and mitochondrial respiratory capacity [53]. This was concluded after a significant increase in mitophagy-related gene (Pink1, Park2, Park7, and Bnip3) and autophagy-related gene (Becn1) was observed in the proximal forelimb muscle of 13-week-old mdx mouse model after 10 weeks of UroA (50 mg/kg/day) supplementation. This observation was confirmed by the increase in the mitophagy markers (Phospho-S65-ubiquitin, Parkin, and BNIP3) in skeletal muscle of UroA-treated mdx mice. UroA was also found to increase the mitochondrial content via the increase in the mtDNA/nDNA ratio. Furthermore, the increase in ATP and citrate synthase activity observed by the study are indicative of a more metabolically active mitochondrial. As the impairment in functional muscle stem cells (MuSCs) is also known to worsen the DMD disease [53,91], UroA supplementation on the mdx models was shown to have increased MuSCs and reverse the reduction in the expression of mitophagy marker (Pink 1 and Park 3) in the disease state. Interestingly, enhanced muscle regeneration was also proposed after observation of increased in average cross-sectional area (CSA) of muscle fibres and embryonic myosin heavy chain expression in cardiotoxin-damaged mdx models with transplanted MuSCs from UroA-treated (50 mg/kg/day for 10 weeks) 20-month-old C57BL/6JRi mice. The study also showed UroA (50 mg/kg/day) supplementation reduced fibrosis of diaphragm and cardiac muscle in mdx mice after 10 weeks of supplementation, which is a common cause of death among young DMD patients in the late-disease progression state [92]. Prevention of premature death (*p* ≤ 0.05) in a 5-week-old mdx/Utr^−/−^ (DKO) mice model was also observed. Thus, the study indicated the mitophagy properties of UroA may be a suitable treatment or adjuvant to combat DMD [53].

#### 3.3.9. Osteogenic Activity


*Urolithin A*


Tao et al. (2021) investigated the osteogenic properties of UroA on ovariectomy-induced osteoporosis in the C57BL/6 mice model group [55]. The three-dimensional (3D) image reconstruction showed that UroA administration significantly reduced the extensive trabecular bone loss in the distal femur compared to the OVX control mice group with ovariectomy-induced osteoporosis. At 10 mg/kg, UroA treatment increased the bone mineral density (BMD) (0.088 ± 0.007 g/cm^3^ vs. 0.052 ± 0.005 g/cm^3^) and decreased trabecular separation (Tb.Sp) (0.368 ± 0.015 mm vs. 0.555 ± 0.006 g/cm^3^) compared with the corresponding values in the OVX control group. Histological analysis revealed the surface osteoclast number in the cancellous bone of the distal femur was dramatically lower in the UroA-treated groups than the OVX control group. Fluorescence intensity of NFATc1, a master transcriptional regulator of osteoclastogenesis, was shown to have decreased by UroA treatments. CTX-1, a biomarker for assessing the rate of bone turnover, was found to decrease in a concentration-dependent manner by the treatment of UroA [55,93]. The signalling pathway RANK/RANKL/OPG is responsible for the activation and differentiation of osteoclast [94]. RANKL, a ligand for the receptor RANK, is responsible for the activation and differentiation of osteoclast. RANKL may bind to osteoprotegerin (OPG), preventing the binding of RANKL to RANK, which is essential in preventing bone resorption [94]. Tao et al. revealed that RANKL/OPG ratio was significantly decreased in the treatment of 10 mg/kg and 20 mg/kg UroA compared to the OVX control group. These data suggested that UroA was capable of the suppression of osteoclast activation, and it acted as a prophylaxis against osteoporosis.

The in vivo osteoblast proliferation properties of UroA were studied by Liu et al. (2022) in the induced middle femoral defect mice model [56]. Serum osteocalcin level of the UroA-treated mice group (7.84 ± 0.19 pg/mL) was higher than the non-treated bone defect mice group (2.93 ± 0.13 pg/mL). The serum CTX level was reported to be significantly reduced in the UroA-treated group (61.7 ± 3.1 ng/mL) compared to the non-treated bone defect group (152.9 ± 5.2 ng/mL). Histological analysis showed that UroA treatment promoted new bone formation, angiogenesis, and increased the histopathologic score. In addition, the UroA-treated group showed significant improvement for the bone defect score and enhancement to the femoral BMD compared to the non-treated bone defect mice group. Wnt3a and GSK3β levels were significantly upregulated in the UroA-treated group compared to the bone defect and sham mice group. These results showed that UroA stimulates the transcription of osteogenic markers, contributing to the osteogenic differentiation of osteoblasts.

#### 3.3.10. Radioprotective Activity


*Urolithin A*


In the radiotherapy treatment for gastrointestinal-related cancer, the most common side effect is intestinal damage. Ionising radiation destroys the physiological structure of the intestine and causes life-threatening damage to the intestine [95]. Zhang et al. (2021) investigated the radioprotective activity of UroA on irradiated C57BL/6 mice model group as pre- and post-treatment via intraperitoneal injection [60]. The study revealed that UroA-treated group had the highest survival rate (>5 days) at a dosage of 2 mg/kg (70% of the irradiated (IR) mice in 2 mg/kg UroA-treated group) among all the UroA-treated group (0.4, 2, and 10 mg/kg). Hence, 2 mg/kg UroA was selected for the intestinal analysis. The UroA-treated group was shown to be present with more survival crypts in the small intestine compared to the control group, while restoring the loss of villi height reduced by the 9.0 Gy total body irradiation (TBI). Besides that, Axin2, a protein of a downstream molecule of β-catenin that is essential to cell proliferation and differentiation, was also restored by UroA treatment [60,96]. Ki67, a cellular marker associated with cell proliferation, was also restored by UroA treatment as UroA promotes crypts regeneration [60,97]. Zhang et al. also reported that UroA treatment alleviated the increase of 8-OHdG, a biomarker of oxidative stress and carcinogenesis [60,98]. In the presence of DNA injury, p53 protein responds by triggering apoptosis or inducing a cell cycle arrest. UroA treatment was shown to markedly restore the overexpression of p53 back to normal level. Irradiation of 9.0 Gy caused the upregulation of caspase8 and caspase3, which were downregulated by UroA treatment. This indicates that UroA could ameliorate radiation-induced intestinal damage by modulating the p53-mediated apoptosis pathway [60].

### 3.4. Gene Expressions Modulated by Uro

All evidence points to the potential of UroA at regulating the autophagy/mitophagy pathway to ameliorate metabolic disorders and ageing diseases. Autophagy, in Greek “the eating of oneself”, as described by Kelekar (2006), refers to a major intracellular pathway that is evolutionarily conserved, regulated turnover of cellular constituents that occurs during development, as a response to stress [99]. It degrades and recycles long-lived proteins and cytoplasmic organelles. Mitophagy is the selective autophagic process for mitochondria, where it prevents cellular degeneration caused by the accumulation of dysfunctional mitochondria and promotes the turnover of mitochondria [100].

To validate the potential therapeutic effects of urolithin(s), it is essential to elucidate the mechanism behind the regulation of gene expressions and signalling pathways by urolithin(s) that is responsible for the mitophagy/autophagy. UroA supplementation has shown to normalise mitochondrial morphology, stimulate autophagy, and mitophagy, and regulate inflammatory responses in various mice and rat models [28,29,30,31,32,33,34,35,36,37,38,39,40,41,42,43,44,45,46,47,48,49,50,51,52,53,61,62,63,64,65,68]. The gene expressions regulated by urolithin(s) categorised based on the origins are shown in Table 2.

### 3.5. Autophagy and Mitochondrial Biogenesis Signalling Pathways Modulated by UroA

In short, signal transduction pathway can be described as the binding of extracellular signalling molecules and ligands (urolithin), to receptors located on the cell surface, leading to a cascade of chemical reactions that invoke a response [101]. Ergo, for urolithin(s) to exert mitochondrial biogenesis and its autophagic properties, it is vital to discern the signalling pathway involved in the mechanistic action of urolithin(s).

Sirt1 pathway

The upregulation of miR-34a is found in various ageing-related diseases and it may be responsible for the initiation of the apoptotic signalling pathway [102]. It was revealed in a study by Chen et al. (2019) that the modulation of Sirt1 signalling also inhibits the expression of miR-34a in the hippocampal tissue [37]. As such, UroA was shown to upregulate Sirt1 and downregulate p53/p21. P53/p21 has been established to be associated with the regulation of cell cycle, DNA repair, apoptosis, and other important cellular functions. This resulted in the inhibition of apoptosis and promoting autophagy. Furthermore, Sirt1 was proposed to regulate autophagy by manipulating mammalian target of rapamycin (mTOR) and AMP-activated protein kinase (AMPK) activity via the deacetylation of liver kinase B1 (LKB1) [103]. On the other hand, AMPK was also able to activate Sirt1 [103]. Besides that, it was also shown that the Sirt1 helps to regulate autophagic degradation via the FOXO/p53 signalling pathway [104]. Sirt1 regulates autophagy indirectly through the deacetylation of the FOXO1 and FOXO3 [103]. Nevertheless, there also have been reports on Sirt1 interactions with autophagic genes *Atg5* and *Atg7*, in forming molecular complexes, which are associated with autophagic machinery.

AMPK pathway

Gong et al. (2019) investigated the AMPK signalling pathway involvement in the mechanistic action of UroA to exert its autophagic properties in the cortex and hippocampus of APP/PS1 mice. Increased expression of p-AMPK and decreased expression of p-P65NF_K_B, p-P38 MAPK, Bace1, and APP were reported in the UroA-treated APP/PS1 animal model group [43]. AMPK activation increased the autophagic signalling by activation of the UNC-51-like kinase (ULK) 1, phosphorylates mitochondrial fission factor (MFF), activated mitochondrial biogenesis (peroxisome proliferator-activated receptor gamma coactivator-1 alpha (PCG-1α)), and modulated transcriptional regulators of autophagy and lysosomal genes accordingly [105,106]. Another transcriptional regulation of autophagy by AMPK was through the phosphorylation of the FOX family of transcription factors. The phosphorylation of FOXO3 activates the autophagy genes. This results in the activation of autophagy genes [105]. Whenever AMPK is activated, mTOR will be suppressed. mTOR subsequently dephosphorylate FOXK1 and FOXK2. FOXK1 and FOXK2 function as with competitors of FOXO3 for the binding of the same genomic regulatory site of autophagy genes [105].

PI3K/AKT/mTOR pathway

Interestingly, mTOR pathway regulates the transcriptional factor families, such as FOXO, FOXA, NRF, NF-_K_B, SREBPs, and TFEB. mTOR functioned in a negative feedback manner against the AMPK pathway [107]. However, it has become particularly clear there is involvement of antagonistic pleiotropy nature. Evidence reported by Tuohetaerbaike et al. (2020) showed that UroA increased p-AKT and p-mTORC1 levels in the pancreas of the C57BL/6 mice model group [51]. There has been a report on the mTOR complex 1 (mTORC1) signalling in the maintenance of postnatal β cell mass for the regulation of apoptosis, size, and proliferation [108]. Its impairment can lead to failure in autophagy, apoptosis, and other physiological defects [108]. The cascade signalling was described as follows: UroA increased the AKT/mTOR phosphorylation and the activation of PI3K dependent-AKT phosphorylates and inhibits tuberous sclerosis complex 2 (TSC2), allowing Rheb to activate mTOR1 [109].

### 3.6. Implications

Hitherto, UroA was the most widely studied, since it was discovered in 2005 [19]. Clinically, UroA has shown to improve mitochondrial and cellular health in elderly individuals, indicating UroA’s ability to reduce the decline in mitochondrial function that accompanies ageing and to promote healthy muscle function [23]. Recent randomised control clinical trials also showed that UroA is safe and well tolerated, as well as being beneficial for muscle endurance and mitochondrial health [22]. The mitophagy, mitochondrial biogenesis, and autophagy characteristics of UroA attributes to its numerous potential therapeutic effects (i.e., musculoskeletal protection, neuroprotection, and anti-metabolic effects) as reported in the in vivo studies [35,36,37,43,47,51,53,58,87]. Based on the evidence gathered, it can be postulated that UroA binds mainly to the pharmacological targets involved in the cellular senescence and longevity regulating pathways.

Nonetheless, evidence reported by these in vivo studies might not be an accurate representation of the nutritional benefits that ET consumption provides. In the aspect of therapeutic effects, there is a distinctive difference between the nutritional bioactivity of ET-rich food and pharmacological activity of Uro. These compounds are produced by gut microbes in the colon after consumption of ET-rich food, and its variation is highly associated with individual gut microbiota composition (Uro metabotypes). The highly bioavailable Uro is absorbed mainly in the colonic mucosa, where it is subjected to phase II metabolism. This results in the formation of glucuronic and sulphate conjugates of urolithins. These Uro conjugates reach the plasma and systemic tissues within three to four days and exert their bioactivity [24]. Recent clinical trials involving the consumption of ET-rich food recorded up to 35 µmol/L glucuronic Uro conjugates, while its aglycones were in the range of 0.005 µmol/L [110,111]. Pharmacokinetic studies on urolithins clearly showed that urolithin glucuronides were the dominant metabolites detected in tissue, plasma, and urine [7]. Piwowarski et al. (2017), in their studies with comparative studies of Uro and their phase II metabolites on macrophages and neutrophil functions, showed that there was a selective activation of Uro glucuronides at the inflammation and infection sites [112]. Upon the stimulation by *N*-formylmethionine-leucyl-phenylalanine, β-glucuronidases were released by neutrophils from its azurophilic granules. Uro A, UroB, and Iso-Uro A glucuronides were shown to be cleaved by β-glucuronidases, resulting in increased concentration of pharmacologically active Uro aglycones. The predominant in vivo experiments in this study, however, were conducted using Uro aglycones, and were administered at doses not attainable through ET-rich food consumption. In consideration of the above-mentioned facts, it is imperative that one takes into account the route of administration, dosage of Uro, pharmacological targets, and metabotypes when considering Uro or ET for its pharmacological or nutritional application.

### 3.7. Strength and Limitations

The results from this systematic review were summarised and qualitatively synthesised from forty-one individual studies that investigated the pharmacological properties, modulation of gene expression, and signalling pathway involved in the mechanistic action of urolithin(s). This review provides a comprehensive overview of the latest research from January 2011 to December 2021. Thus, the results summarised within this systematic review provide an overall representation of the bioactive properties of urolithin(s) studies thus far. In addition, this systematic review follows the Preferred Reporting Items for Systematic Reviews and Meta-Analysis (PRISMA) to provide a systematic yet extensive summary of the present research [25]. Eleven out of thirty-two studies utilised rats as the animal models, while the majority of the studies included diverse strains of mice models. Thus, there is a good amount of homogeneity of the experimental group of mice, as well as a relatively accurate and comprehensive summary of the latest available literature. Moreover, this systematic review addresses a crucial gap in the current literature surrounding the pharmacological activities of urolithin(s), as well as the understanding of the genes and mitophagy pathway involved. Nonetheless, there were various restrictions and drawbacks that should be noted for this review. There exist several heterogeneities in this review: language search was limited to English, diverse outcome measurements, lack of diversity in the types of Uro, formulations of the administered Uro(s), and animal models. Among the studies reviewed, it was noted that UroA and UroB were widely studied, possibly due to it being most commonly found at higher concentrations in the circulatory system compared to urolithin C (UroC) and D (UroD) [14]. In addition to that, the studies conducted with UroC and UroD were mostly in vitro, which did not fit into the selection criteria, and thus, they were excluded from this study. Furthermore, there is a lack of reported analysis on the gene expressions modulated by urolithin(s), and the signalling pathway involved. Cell-based studies were also excluded in this systematic review. This might due to the exclusion of gene expression and metabolic pathway studies in cell-based studies. The purpose of a systematic review is to provide an overview of the existing evidence, and not critically appraise the quality of the studies. Hence, this review only covers qualitative synthesis of an in-depth overview of the existing evidence based on the selected studies, factoring out their intrinsic qualities. To the best of our ability, SYRCLE’s risk of bias tool was employed to assess the intrinsic qualities of the included studies.

### 3.8. Future Research

Although the present study aims to provide an overview of the Uro’ bioactivities, it is limited by the scarcity of in vivo evidence of the bioactivities of Uro C and D, given that previous in vitro literature has proven their potential bioactivities, including an anti-metabolic and anti-tumour effect [14,113,114]. Thus, future in vivo research is needed to explore the relevant bioactivities of other Uro. As the current studies focused on individual types of Uro, it remains unknown whether administration of mixtures of different Urolithins may provide a synergistic or counteractive effect [113]. Additionally, it shall be noted that these in vivo studies discussed either administered Uro as a preventive or intervention treatment. Those that utilised Uro as a prophylaxis treatment mainly involved in vivo studies demonstrating nephroprotective and cardioprotective effects against ischemic reperfusion injury in kidney observed during kidney transplant and in acute myocardial infarction [32,46,50,59]. The role of Uro as a prophylaxis therapy in ischemic reperfusion injury, which is high in oxidative stress, was explained as an intervention to help the body to get into a preconditioned high anti-oxidants state and preserve the organ function [115]. Hence, future studies can explore the use of Uro either as a prophylaxis or therapeutic agent based on the disease pathophysiology and desired Uro properties. Furthermore, few in vivo studies have begun to explore the use of modified Uro in order to increase Uro’s bioavailability, as it is prone to hydrolysis by the digestive enzymes and acidic environment [33]. The modifications of the chemical structure, such as UAS03 (a non-hydrolysable derivative), were done by using ligands conjugated nano-delivery method (P2Ns-GA encapsulation) and methylation to increase the bioavailability and potency of Urolithins [33,49,68]. As such, more research can be performed to explore the ways to increase oral bioavailability and provide a comprehensive pharmacology, pharmacokinetics, and toxicology evaluation of the derivatives. Lastly, the limited number of clinical trials assessing the therapeutic effects of urolithins and the vast range of dosage and administration method of Uro used in vivo warrants more in-depth research to be performed to provide a more promising and conclusive result of the usage of Uro as a therapeutic or preventive intervention.

## 4. Conclusions

This review provides scientific justification for the potential development of Uro as a potent natural mitophagy inducer and therapeutic agent for age-associated disease. Overall, this systematic review methodically discusses and describes the prospect of the therapeutic aspect of Uro A and B, its synthetic analogue UAS03, and methylated UroA, and provides insights into potential future research. It also provides a summarised overview on the gene expressions, autophagy, and mitochondrial biogenesis signalling pathway modulated by Uro. The studies on the signalling pathway and genes involved in the mechanistic action of Uro should be studied extensively in future investigations to elucidate the relation between mitophagy and the therapeutic potential of Uro.

## Figures and Tables

**Figure 1 nutrients-14-03494-f001:**
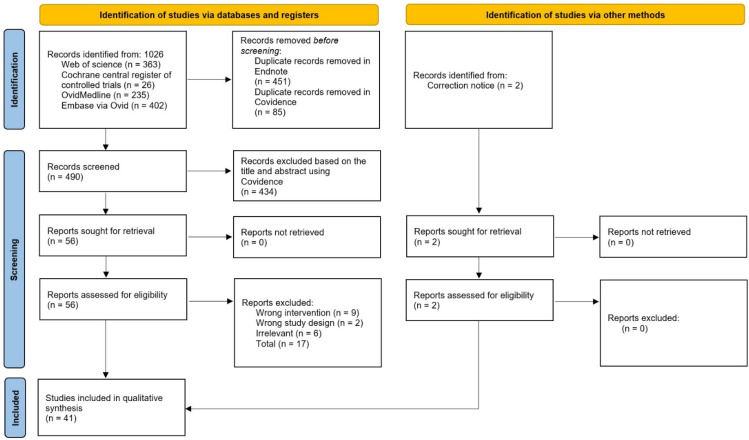
PRISMA flow diagram for the article selection process.

**Figure 2 nutrients-14-03494-f002:**
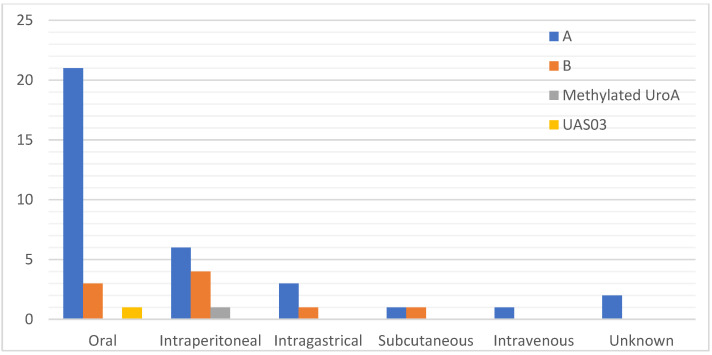
Frequency of the mode of administration route used in the included studies categorised by the type of Uro.

**Figure 3 nutrients-14-03494-f003:**
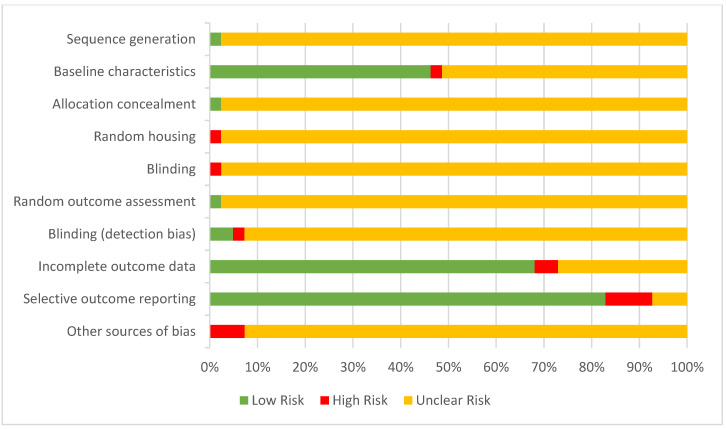
Risk of bias assessment using SYCRLE’s risk of bias tool.

**Table 1 nutrients-14-03494-t001:** Types of urolithin, bioactive properties, intervention type, administration route, dosage, frequency, and duration of the administration.

Uro	Bioactive Properties	Intervention Type	Route of Administration	Dosage	Frequency	Duration	Reference(s)
**A**	Anti-tumour	Treatment	Oral	50 mg/kg	5 days per week	4–5 weeks	[28]
**A**	Neuroprotective	Treatment	Oral	10 mg/kg	Daily	1 month	[29]
				25 mg/kg			
				50 mg/kg			
**A**	Anti-metabolic	Treatment	Intravenous	20 µg	Daily	12 weeks	[30]
**A**	Anti-metabolic	Treatment	Oral	4 mg/kg EA	N/R	8 weeks	[31]
				4 mg/kg UroA			
				4 mg/kg EA +4 mg/kg UroA			
**A**	Nephroprotective	Preventive	N/R	50 mg/kg	3 days and 30 min pre-surgery	3 days	[32]
**A, UAS03**	Anti-inflammatory	Treatment	Oral	20 mg/kg	12 hourly	3 days	[33]
		Preventive			6 hourly pre-induction	1 day	
		Treatment			4th and 6th day post induction	6 days	
**A**	Anti-inflammatory	Treatment	Oral	40 mg/kg	Once	8 h	[34]
**B**	Anti-metabolic	Treatment	Oral	10 mg/kg	Daily	14 days	[61]
**A**	Musculoskeletal protective	Treatment	Oral	50 mg/kg	Daily	34 weeks	[35]
				50 mg/kg		6 weeks	
				25 mg/kg		24 weeks	
**B**	Anti-tumour	Treatment	Intraperitoneal	40 mg/kg	Once every 2 days	1 month	[62]
		Treatment	Subcutaneous	40 mg/kg	Once every 2 days	1 month	
**A**	Musculoskeletal protective	Treatment	Intragastric	10 mg/kg	N/R	16 weeks	[36]
**A**	Neuroprotective	Treatment	Oral and subcutaneous	150 mg/kg	Daily	8 weeks	[37]
				100 mg/kg			
				50 mg/kg			
**A**	Hepatoprotective, nephroprotective	Treatment	Oral	150 mg/kg	Daily	8 weeks	[38]
				100 mg/kg			
				50 mg/kg			
**A**	Nephroprotective	Treatment	Oral	50 mg/kg	Daily	1 week	[39]
**A**	Musculoskeletal protective	Treatment	Oral	25 mg/kg	Daily	28 days	[40]
**A**	Nephroprotective	Treatment	Oral	100 mg/kg	N/R	5 days	[41]
**A**	Anti-inflammatory	Treatment	Oral	0.114 mg/kg	Daily	6 days	[42]
**A**	Neuroprotective	Treatment	Oral	300 mg/kg	Daily	2 weeks	[43]
**Methylated UroA**	Anti-tumour	Treatment	Intraperitoneal	20 mg/kg	Once every 4 days	28 days	[68]
				40 mg/kg			
				80 mg/kg			
**A**	Neuroprotective	Treatment	Intraperitoneal	30 µg	N/R	39 days	[44]
**A**	Anti-metabolic	Treatment	Oral	50 mg/kg	N/R	56 days	[45]
**A**	Myocardial protective	Preventive	Intraperitoneal	1 mg/kg	24 h and 1 h pre-induction	1 day	[46]
**B**	Myocardial protective	Treatment	Intraperitoneal	2.5 mg/kg	Daily	2 weeks	[63]
				5 mg/kg			
**A**	Musculoskeletal protective	Treatment	Oral	25 mg/kg	Daily	28 days	[47]
**B**	Myocardial protective	Treatment	Intraperitoneal	0.7 mg/kg	0, 24, and 48 h	2 days	[64]
**A**	Neuroprotective	Treatment	Oral	200 mg/kg	Daily	90 days	[48]
				200 mg/kg		30 days	
**A**	Nephroprotective	Treatment	Oral	50 mg/kg	3 times per week	19 days	[49]
**A**	Myocardial protective	Treatment	Intraperitoneal	30 mg/kg	Once		[50]
**A**	Anti-metabolic	Treatment	Intragastric	50 mg/kg	Daily	98 days	[51]
**A, B**	Anti-metabolic	Treatment	Intraperitoneal	2.5 mg/kg	4 times per week	28 days	[52]
**B**	Nephroprotective	Treatment	Oral	20 mg/kg	Daily	21 days	[65]
				40 mg/kg			
				80 mg/kg			
**A**	Musculoskeletal protective	Treatment	Oral	50 mg/kg	Daily	70 days	[53]
**B**	Anti-inflammatory	Treatment	Oral	150 mg/kg	Daily	N/R	[66]
				100 mg/kg			
				50 mg/kg			
		Treatment		450 mg/kg	Daily	28 days	
				300 mg/kg			
				150 mg/kg			
		Treatment		150 mg/kg	Daily	8 weeks	
**B**	Neuroprotective, anti-inflammatory	Treatment	Intragastric	150 mg/kg	Daily	8 weeks	[67]
				100 mg/kg			
				50 mg/kg			
		Treatment	Oral	150 mg/kg	Daily	8 weeks	
**A**	Anti-metabolic	Treatment	Oral	50 mg/kg	Twice per day	8 weeks	[54]
**A**	Osteogenic activity, anti-inflammatory	Treatment	Intragastric	10 mg/kg	Daily	8 weeks	[55]
				20 mg/kg			
**A**	Osteogenic activity	Treatment	N/R	Unknown	N/R	N/R	[56]
**A**	Myocardial protective	Treatment	Intraperitoneal	2.5 mg/kg	Daily	8 weeks	[57]
**A**	Musculoskeletal protective	Treatment	Oral	25 mg/kg	Daily	4 weeks	[58]
**A**	Nephroprotective	Preventive	Oral	20 mg/kg	Daily	1 week	[59]
				50 mg/kg			
				100 mg/kg			
**A**	Radioprotective activity	Preventive	Intraperitoneal	0.4 mg/kg	48 h, 24 h, and 1 h prior to and 24 h after induction	3 days	[60]
				2 mg/kg			
				10 mg/kg			
		Preventive		2 mg/kg	48 h, 24 h, and 1 h prior to and 24 h after induction	3 days	

N/R: Not reported.

**Table 2 nutrients-14-03494-t002:** Gene expressions regulated by urolithin(s) reported in the included in vivo studies.

Urolithin(s)	Origins	Genes Upregulated	Genes Downregulated	Reference(s)
**A**	Liver	*Cpt1 **	N/R	[30]
		*Sirt1 **		
		*Il1b **		
		*Sod1 ***		
		*Sod2 ***		
		*Mfn2 **	N/R	[31]
		*Prkn **		
		*Pink1 **		
	Skeletal muscle	*Mfn2* *	N/R	[31]
		*Prkn **		
		*Pink1 **		
	Colon	*Cldn4 **	N/R	[33]
	Macrophages	*Nrf2 **	N/R	
		*HO1 **		
	Gastrocnemius muscles	*Becn1*	N/R	[35]
		*Ulk1*		
		*Pik3c3 ****		
		*Atg8l*		
		*Park2 ****		
		*p62*		
		*Atg5*		
		*Atg7*		
		*Atg12*		
		*Lc3b*		
		*Lamp2*		
	Murine vastus lateralis muscle	*Gata6 **	N/R	[36]
		*Hgf **		
		*Nrp1 **		
		*Dab2 **		
		*Cyr61 **		
		*Vegfa **		
		*vWF **		
		*Vegfr2 **		
		*Pecam1 **		
		*Gata2 **		
		*CD105 **		
		*Tnc **		
	Hippocampal	*Gabra2*	*Asph*	[48]
		*Rap1gap2*	*Nrxn3*	
		*Lrrtm4*	*Tnik*	
		*Slitrk1*	*Grin2a*	
		*Lin7a*		
		*ll1rapi2*		
		*Lgi2*		
		*Arpp21*		
		*N28178*		
		*Kcnv1*		
		*Sncb*		
		*Ache*		
		*Grm1*		
	Renal cortex	N/R	*Mt1 ****	[49]
			*Txnrd1* **	
			*Srxn1* ***	
			*Cdkn1* **	
			*Atf3* ***	
			*Trp53inp1* ****	
	Cardiac ventricular tissue	*Art5 **	*N/R*	[50]
		*mtDNAj **		
		*ClpP **		
		*LonP1 **		
		*CHOP **		
		*Hsp10 **		
		*Hsp60 **		
	Hindlimb muscles	*Pink1 *****	N/R	[53]
		*Park2 *****		
		*Bnip3*		
		*Sqstm1*		
		*Becn1*		
	Bone tissue	N/R	*Sox9 ***	[56]
			*Col2a1 ***	
			*Runx2 ***	
			*MMP13 ***	
			*Osterix ***	
**B**	Cardiac	N/R	*STAT3*	[63]
**UAS03**	Colon	*Cldn4 **	N/R	[33]
	Macrophages	*Nrf2 **		
		*HO1**		

Statistically significant compared to model group: * *p* < 0.05; ** *p* < 0.01; *** *p* < 0.001; **** *p* < 0.0001; N/R: Not reported.

## Data Availability

Not applicable.

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
