# Peer review of "The Therapeutic Relevance of Urolithins, Intestinal Metabolites of Ellagitannin-Rich Food: A Systematic Review of In Vivo Studies"

_nutrients, 2022, doi:10.3390/nu14173494_

Round 1

Reviewer 1 Report

The authors collate and summarize the available data on urolithin therapy, highlighting its various functions when applied as a drug, providing great help for future research in related directions, and for urolithin A in anti-aging potential provides a scientific basis. Some suggestions for context are as follows.

1.    Due to the title "The Therapeutic Relevance of Urolithins, an Intestinal Metabolite of Ellagitannin-rich Food", an introduction of Ellagitannin-rich Food should be given. This will enrich the background presentation of the article.

2.    The data search and literature analysis methods used in the article should be explained in the introduction part, so that it is easier for non-professionals in the field to understand.

3.    Given the focus of the article, some new articles related to urolithin bioavailability and antioxidant activity should also be referred to.

Author Response

Reviewer 1

No

Comments

Rebuttal

Action Taken

Lines

1

Due to the title "The Therapeutic Relevance of Urolithins, an Intestinal Metabolite of Ellagitannin-rich Food", an introduction of Ellagitannin-rich Food should be given. This will enrich the background presentation of the article.

We thank the reviewer for this comment

A short background for ellagitannin-rich food has been added to the introduction.

“ETs are commonly found in foods such as fruits (pomegranate, rambutan, longan, and guava), nuts (chestnuts, pecans, walnuts, and almonds), and berries (blackberries, strawberries, cloudberries, and raspberries)”

Line 35 – 37

2

The data search and literature analysis methods used in the article should be explained in the introduction part, so that it is easier for non-professionals in the field to understand.

The data search and literature analysis methods were thoroughly described and detailed in the methodology section following the method of Reporting Items for Systematic reviews and Meta-Analyses (PRISMA) [1].

Line 84 - 127

3

Given the focus of the article, some new articles related to urolithin bioavailability and antioxidant activity should also be referred to.

The objective of this systematic review is to provide an overview on the potential regimen, therapeutic effect, and the proposed mechanisms of autophagy and mitochondrial biogenesis of Uro. Bioavailability was not an aim of this study.

Antioxidant ability of urolithins is described throughout the paper, playing a role in its therapeutic ability.

The authors have added new evidence of urolithin bioavailability in the introduction (as reference).

Line 61

Reviewer 2 Report

The paper submitted by Tow et al. is a comprehensive review on the therapeutic relevance of urolithins. The manuscript os well prepared and interesting. however some corrections can be done to improve its scientific soundness.  

1) deeper search for literature should be performed - only few studies of Espin and Barberan are cited, while works of Piwowarski et al. or Bauer el al. were omitted 

2) more info on the fate of ellagitannins and urolithins after oral ingestion should be given

3) please discuss in details the II phase metabolism of urolithins after being processed in the gut and influence of II phase metabolism on the bioactivity of Urolithins should be mentioned

Author Response

Reviewer 2

No

Comments

Rebuttal

Action Taken

Lines

1

deeper search for literature should be performed - only few studies of Espin and Barberan are cited, while works of Piwowarski et al. or Bauer el al. were omitted 

The authors carried out the systematic review with the objective of synthesising publication on the potential regimen, therapeutic effect, and the proposed autophagy and mitochondrial biogenesis mechanisms of Uro following the method of Preferred Reporting Items for Systematic reviews and Meta-Analyses. The publications of Piwowarski et al. and Bauer et al. were not captured and the mentioned studies of Espin et al. and Baberan et al. were not captured in our search methodology as it did not meet the selection criteria.

We agree that these authors have contributed significantly to this area, we have since captured some of their work in the newly added sub-section 3.6: Implications.

Line 869 – 893

2

more info on the fate of ellagitannins and urolithins after oral ingestion should be given

Thank you very much for the comments.

The authors have included details of the fate of ellagitannins and urolithins after oral ingestion in the 3.6: Implications.

Line 872 – 878

3

please discuss in details the II phase metabolism of urolithins after being processed in the gut and influence of II phase metabolism on the bioactivity of Urolithins should be mentioned

Thank you very much for the comments.

The authors have included details of the fate of ellagitannins and urolithins after oral digestion in the 3.6 Implications.

Line 869 – 893

Reviewer 3 Report

In this manuscript, Tow et al reviewed the therapeutic potentials of Uro A and B by focusing on in vivo studies. This review is comprehensive and a substantial work was made to summarize the reviewed studies in table 1 and S1. These tables are valuable.

This review is mostly descriptive and therefore it sometimes sounds like a list / inventory of biological effects mediated by Uro. A common thread between the different sections is lacking sometimes. Nevertheless,  this manuscript will be useful for scientists that address the in vivo pharmacology of Uro.

General comments:

The basic rationale for investigating the therapeutic and/or pharmacological properties of Uro is that consumption of ET-rich foods confers health benefits. However, most in vivo studies report Uro dosages or routes of administration that likely result in circulating Uro concentrations high above the ones that can be reached following food consumption. In addition, as far as I know, following food consumption, Uro are mostly concentrated in colon rather that in the blood stream. Therefore, it is hard to relate the bioactivity and mechanism of action of ET-rich food with most of the in vivo studies that are reviewed here. I believe these points should be more emphasized somewhere in the manuscript.

Therefore, I think the authors should clearly make the difference between the nutritional bioactivity of ET foods (that leads to low circulating concentrations of Uro) and the pharmacological effects of Uro (following administration of high doses of Uro not attainable with food consumption) from the introduction.

The authors made a substantial work to summarize the Uro dosages used in the studies. This is very useful for scientists that would like to start adressing the in vivo effect of Uro. From the studies reviewed by the authors, I would like to know whether it is possible to estimate the average Uro dosage that is likely to have a pharmacological effect in vivo if any. Also do the authors have any idea of the circulating Uro concentrations that result from these pharmacological treatments ?

1- Title: please check the grammar of the title. Also, the title wording should be more focused on Uro A and B as Uro C and D are almost not reviewed owing to the lack of in vivo studies. In addition, the title should include the wording "in vivo" somewhere.

2- Abstract: please specifiy that Uro A and B (or their derivatives) are the only Uro that are highlighted fromthis systematic review

3- introduction:

- lines 47-51 are confusing. Could you please reformulate ?

- explain why most in vivo studies involved Uro A and B rather that C and D. Is this becauses Uro A and B are found at higher circulating concentrations ?

-Table 1: it is very useful. I think the reference #52 should be excluded from the review due to the lack of information regarding the treatment

- Table S1: is this table mentioned in the text ?

4- result section: general remarks

- I would remove all the statistics data (p). It is useless and irrelevant since the reader has no access to the experimental design and the tests that were run.

- whenever possible, when you describe a mechanism/signaling cascade, please specify in which tissue, brain structure etc. it has been investigated. For example, line 254 (where the anti oxidant activities are increased ?), all the data from reference 63 was performed in the hippocampus ?

5- section 3.3.4

line 416: "Study conducted by Yang et al. (2020) revealed that UroA is a potent blood glucose, adi-416 ponectin, and mitochondrial dynamic regulator compared to EA" . This is too vague. What does regulator mean ? increase, decrease, improve, deteriorate ? Please avoid this term throughout your manuscript. It is better to specify.

6- section 3.5 lines 803-807: it is stated that urolithins are ligands of cell surface receptors (= pharmacological targets of Uro). Are Uro able to cross the cell plasma membrane ? Also, It would be of great interests if the manuscript could also mentioned whenever possible the pharmacological targets of Uro. In my opinion this is as interesting as the basic description of the signaling pathways that are triggered by Uro.  The determination of the pharmacological targets of Uro is an important perspective of the field. You can find in the litterature studies that adress this question.

7- conclusion: I am not sure that the first paragraph of this conclusion actually belongs to the conclusion as it is focused on the effect of UroA mitochondria biology.

Author Response

Reviewer 3

No

Comments

Rebuttal

Action Taken

Lines

General Comments

The basic rationale for investigating the therapeutic and/or pharmacological properties of Uro is that consumption of ET-rich foods confers health benefits. However, most in vivo studies report Uro dosages or routes of administration that likely result in circulating Uro concentrations high above the ones that can be reached following food consumption. In addition, as far as I know, following food consumption, Uro are mostly concentrated in colon rather that in the blood stream. Therefore, it is hard to relate the bioactivity and mechanism of action of ET-rich food with most of the in vivo studies that are reviewed here. I believe these points should be more emphasized somewhere in the manuscript.

Thank you for the comment, and the authors agree with the suggestion. We have since added a section “Section 3.6: Implications” to describe the fate of ellagitannins and urolithins after oral ingestion. The section also discusses the phase II metabolism of urolithins after being processed in the gut and the influence of phase II metabolism on the bioactivity of urolithins.

Line 869 – 878

Therefore, I think the authors should clearly make the difference between the nutritional bioactivity of ET foods (that leads to low circulating concentrations of Uro) and the pharmacological effects of Uro (following administration of high doses of Uro not attainable with food consumption) from the introduction.

The concern of dosage and route of administration mentioned in the studies we reviewed is discussed in Section 3.6: Implications.

Line 888 –  893

The authors made a substantial work to summarize the Uro dosages used in the studies. This is very useful for scientists that would like to start addressing the in vivo effect of Uro.

From the studies reviewed by the authors, I would like to know whether it is possible to estimate the average Uro dosage that is likely to have a pharmacological effect in vivo if any.

Unfortunately, it is also difficult to estimate the average of Uro dosage that is likely to exhibit pharmacological effect due to the heterogeneities of these studies.

Also do the authors have any idea of the circulating Uro concentrations that result from these pharmacological treatments?

Unfortunately, the studies included did not report on the circulating Uro concentrations that result in the pharmacological treatments.

1

Title: please check the grammar of the title. Also, the title wording should be more focused on Uro A and B as Uro C and D are almost not reviewed owing to the lack of in vivo studies. In addition, the title should include the wording "in vivo" somewhere.

Unfortunately, we cannot specify Uro A and B only, as there’re studies included that assesses its synthetic analogue as well, such as UAS03 and methylated Uro A.

Thank you for the suggestion.

The title has been amended to “The Therapeutic Relevance of Urolithins, Intestinal Metabolites of Ellagitannin-rich Food: A Systematic Review of In Vivo Studies”.

Line 2 – 4

2

Abstract: please specifiy that Uro A and B (or their derivatives) are the only Uro that are highlighted fromthis systematic review

We have specified Uro A, B, and its synthetic analogue in the abstract and conclusion.

Abstract: “The included studies highlighted the neuroprotective, anti-metabolic disorder activity, nephroprotective, myocardial protective, anti-inflammatory, and musculoskeletal protection of urolithin A, B, and its synthetic analogue methylated urolithin A.”

Line 21 – 23

Conclusion: “Overall, this systematic review methodically discusses and describes the prospect of the therapeutic aspect of Uro A, B, its synthetic analogue UAS03 and methylated UroA and provides insights into potential future research.”

Line 958 – 960

3

introduction:

- lines 47-51 are confusing. Could you please reformulate?

Thank you for the suggestion.

We have restructured the sentence.

“Additionally, a recent study showed that individuals with a higher Firmicutes-to-Bacteroidetes ratio (where a low ratio was associated to gut dysbiosis), and a greater abundance of Clostridiales, Ruminococcaceae and Akkermansia muciniphilia are a better UroA producer.”

Line 51 – 54

- explain why most in vivo studies involved Uro A and B rather that C and D. Is this becauses Uro A and B are found at higher circulating concentrations?

UroC and UroD were mostly conducted in in vitro studies. Thus, they are excluded from this review.

We have added a statement in the Section 3.7: Strength and Limitations that discusses the lack of studies on UroC and UroD.

“There exist several heterogeneities in this review: language search limited to English, diverse outcome measurements, lack of diversity in the types of Uro, formulations of the administered Uro(s), and animal models. Among the studies reviewed, it was noted that UroA and UroB were widely studied, possibly due to it being most commonly found at higher concentrations in the circulatory system compared to urolithin C (UroC) and D (UroD). In addition to that, the studies conducted with UroC and UroD were mostly in vitro, which did not fit into the selection criteria, thus they were excluded from this study.”

Line 910 - 917

-Table 1: it is very useful. I think the reference #52 should be excluded from the review due to the lack of information regarding the treatment

As part of the systematic review protocol Preferred Items for Reporting Systematic reviews and Meta-Analyses (PRISMA), the authors are expected to extract and report all data mentioned in the studies, as such we are unable to exclude reference #52

- Table S1: is this table mentioned in the text?

Thank you for the suggestion.

A sentence has been added to 3. Results to reference table s1.

“The extracted data were collated in Table S1.”

Line 134

4

result section: general remarks

- I would remove all the statistics data (p). It is useless and irrelevant since the reader has no access to the experimental design and the tests that were run.

Thank you for the suggestion.

All the statistics data (p value) have been removed from the texts. However, the authors feel that it is essential to keep the statistics data (p value) in the Table S1 and Table 2 for the readers.

- whenever possible, when you describe a mechanism/signaling cascade, please specify in which tissue, brain structure etc. it has been investigated. For example, line 254 (where the anti-oxidant activities are increased?), all the data from reference 63 was performed in the hippocampus?

The author of the original article (reference 63) did not specify the exact anatomical location, only stated brain tissue.

Thank you for the comment.

Location of the investigation has been added to the sentences.

“Moreover, UroB was reported to exert antioxidant properties that increased the level of biochemical indicators such as GSH-Px, SOD, CAT, and T-AOC in the brain tissue of the D-gal induced mice.”

Line 250 – 253

Thank you for the comment.

Location of the investigation has been added to the sentences.

“Gong et al. (2019) investigated the AMPK signalling pathway involvement in the mechanistic action of UroA to exert its autophagic properties in the cortex and hippocampus of APP/PS1 mice.”

Line 826 – 828

5

line 416: "Study conducted by Yang et al. (2020) revealed that UroA is a potent blood glucose, adi-416 ponectin, and mitochondrial dynamic regulator compared to EA”. This is too vague. What does regulator mean? increase, decrease, improve, deteriorate ? Please avoid this term throughout your manuscript. It is better to specify.

Thank you for the suggestion.

An either increase or decrease has been added to each therapeutic effect (etc, potent blood glucose (decrease)).

“Study conducted by Yang et al. (2020 revealed that UroA is a potent blood glucose (decrease), adiponectin (increase), and mitochondrial dynamic (increase) regulator compared EA.”

Line 417 – 419

6

section 3.5 lines 803-807: it is stated that urolithins are ligands of cell surface receptors (= pharmacological targets of Uro). Are Uro able to cross the cell plasma membrane? Also, it would be of great interests if the manuscript could also mention whenever possible the pharmacological targets of Uro. In my opinion this is as interesting as the basic description of the signaling pathways that are triggered by Uro.  The determination of the pharmacological targets of Uro is an important perspective of the field. You can find in the literature studies that adress this question.

Thank you for the comment.

The authors have addressed that Uro has been found in tissues, indicating that Uro are able to cross cell plasma membrane.

The authors have addressed the pharmacological targets of Uro in 3.6: Implications.

Line 852 – 862

7

conclusion: I am not sure that the first paragraph of this conclusion actually belongs to the conclusion as it is focused on the effect of UroA mitochondria biology.

Thank you for the suggestion.

This part has been moved to Section 3.6: Implications.

Line 857 – 867